# OpenReview forum: "Robustly Improving LLM Fairness in Realistic Settings via Interpretability"
_ICLR.cc/2026/Conference — Submitted to ICLR 2026_

### Official Review · Reviewer_Y857 · 2025-10-17

**Soundness:** 3
**Presentation:** 3
**Contribution:** 3
**Rating:** 6
**Confidence:** 3

**Summary:**

This paper demonstrates that simple prompt-based anti-bias mitigations for LLMs, which work well in controlled evaluations, fail when realistic contextual details are introduced to hiring scenarios. The authors show that adding company-specific information, culture descriptions, and selective hiring constraints induces significant racial and gender biases (up to 12% differences in interview rates) across leading commercial and open-source models. To address this, they propose internal bias mitigation using affine concept editing on model activations, which robustly reduces bias to under 2.5% while maintaining model performance.

**Strengths:**

- Important real-world problem: The paper addresses a critical issue as LLMs are increasingly deployed in high-stakes hiring applications with direct impact on people's livelihoods.

- Strong empirical findings: The demonstration that prompt-based mitigations become brittle under realistic conditions is well-documented across multiple models and scenarios.

- Robust internal intervention: The proposed affine concept editing approach shows consistent effectiveness across different contexts.

- Comprehensive evaluation: The experiments cover multiple commercial and open-source models, various contextual conditions, and include both explicit and implicit demographic indicators.

**Weaknesses:**

- Dataset quality issues: The authors acknowledge in Appendix E that 22% of resumes contained unintended demographic indicators, though they claim minimal impact on results.

- Mechanistic clarity and design choices. Directions are estimated from synthetic data and applied at all layers/tokens. The paper doesn’t ablate which layers matter, how many directions per attribute are needed, or compare ACE against other linear methods (e.g., LEACE, NOP, DAS) in this setting. Gemma-3 sensitivity indicates model-specific fragility that deserves deeper analysis.

**Questions:**

- How were company culture blurbs sourced, token-length-matched, and cleaned (e.g., removal of DEI phrases)? Do results persist under length-matched neutral placebos or shuffled company/job pairings?

- ACE application details: which layers carry most of the effect, and does restricting ACE to those layers recover similar mitigation with less capability impact (especially on Gemma-3)?

- How stable are the identified bias directions across different random seeds and prompt orderings?

---

> ### Author Response · Authors · 2025-11-18
>
> We thank the reviewer for their positive assessment and constructive feedback. We address each concern below:
>
> **W1: Dataset quality issues**
>
> As our spot-checks (Appendix E) show minimal impact, we chose to use the exact dataset and evaluation from Veldanda et al. (2023). One of our contributions is demonstrating that existing bias evaluations in the literature, which found no bias with anti-bias prompts, are brittle to additional nuanced context. Using the original dataset without modification allows for a direct comparison to prior work.
>
> **W2: Mechanistic clarity and design choices**
>
> To be clear, we do not view ourselves as proposing a new method. We applied what we consider one of the simplest baseline approach from the interpretability literature: directional ablation following Arditi et al. (https://arxiv.org/abs/2406.11717) and Marshal et al. (https://arxiv.org/abs/2411.09003). The strong performance of this method (always <2.5% bias, usually <1%) with minimal degradation on most models motivated us not to pursue additional methods from the literature.
> We acknowledge the limitations the reviewer identifies. Layer-specific analysis and comparison with alternative methods (LEACE, NOP, DAS) would strengthen the mechanistic understanding. However, our contribution is empirical: demonstrating that a simple internal baseline generalizes robustly across models and realistic contexts where prompt-based methods fail. The Gemma-3 sensitivity is a limitation, and investigating this sensitivity would be valuable future work.
>
> **Q1: Company culture sourcing and cleaning**
>
> Company culture blurbs were manually extracted from public career pages (URLs in Appendix K). In Appendix K, we also include the prompts before and after DEI language filtering. We did not test length-matched neutral placebo or shuffled company/job pairings, though we agree these would be valuable controls for future work.
>
> **Q2: Layer-specific ACE effects**
>
> We chose to follow the methodology of Arditi et al. (2024) by applying the intervention at all layers to minimize the use of additional hyperparameters, as we found this approach yielded strong performance.
>
> **Q3: Stability of bias directions**
>
> Directions were computed from the full Tamkin et al. dataset with no sampling, so there is no seed variation in extraction. We agree that testing stability via subsampling would strengthen robustness claims and view this as valuable future work.

---

> ### Comment · Reviewer_Y857 · 2025-11-25
>
> Thanks for the response. I appreciate the authors' efforts in addressing the reviewers' feedback, and I acknowledge that I have fully read the authors' responses to this and other reviews. While I remain positive about the potential of this work, I find myself in agreement with the other reviewers that the paper would benefit from substantial reframing to address the feedback provided. I believe such revisions would significantly strengthen the contribution and clarity of the work.

---

### Official Review · Reviewer_TqmZ · 2025-10-31

**Soundness:** 1
**Presentation:** 2
**Contribution:** 2
**Rating:** 2
**Confidence:** 4

**Summary:**

This paper investigates biases in LLMs in the context of hiring decisions. The authors first show that prompt-based debiasing techniques can improve model fairness only superficially, while they reduce bias in simple prompt settings, the bias reemerges (by up to 12%) once additional contextual details are introduced into the same task. To address this, the authors propose a representation-level intervention that edits demographic concepts in the model’s internal representations at inference time. This approach aims to directly suppress biased latent associations without retraining. The authors demonstrate that this intervention reduces bias in 4x models while maintaining minimal degradation in general model performance (MMLU).

**Strengths:**

1. Open-source contribution: The authors release their codebase, data and method, which supports transparency and allows for reproducibility and future extensions.

2. Problem relevance: Bias in LLM-based hiring systems is an important and timely issue. The focus on robustness of debiasing under varying context complexity is conceptually interesting.

**Weaknesses:**

1. Overstatement of realism: The paper overclaims the “real-world” nature of its simulated hiring settings. Adding elements like company names, cultural descriptions, or hiring constraints (e.g., “hire the top 10% of candidates”) adds contextual richness, but it does not necessarily make the task realistic. The authors provide no evidence (e.g., comparisons to real hiring data or expert validation) to support that these additions meaningfully increase realism.

2. Prompt debiasing claims conflict with prior work: The finding that prompt-based debiasing is effective contrasts with several prior works that show the opposite (e.g., Rethinking Prompt-based Debiasing in LLMs, Yang et al.; Social Bias Evaluation for LLMs Requires Prompt Variations, Hida et al.). The paper should reconcile these discrepancies or more carefully contextualize its findings in light of this literature.

3. Missing prompt details: The authors mention using four debiasing prompts but fail to disclose their content. This omission significantly limits reproducibility and makes it impossible to assess whether the prompts are reasonable or comparable to prior work.

4. Insufficient baselines: The paper only tests vanilla prompt-based debiasing, ignoring stronger and better-established prompt-debasing baselines such as iterative self-debiasing (Gallegos et al., Self-Debiasing LLMs), implication and self-refinement prompts (Furniturewala et al., Thinking Fair and Slow). Without such comparisons, the conclusion that prompt-based methods are inadequate under context variation is not sufficiently supported.

5. Loose use of interpretability: The paper claims interpretability benefits from its latent editing method but provides no concrete interpretability evidence. E.g., analysis of sparsity, or semantic coherence of modified directions. If interpretability is a claimed contribution, this must be empirically demonstrated.

6. Choice of solution requires justification: the proposed method resembles existing inference-time steering approaches such as FairSteer (Li et al.), which also modulate demographic activation directions but allow for dynamic control of bias levels. The authors should explain why their approach is preferable or complementary to these alternatives.

7. Limited Fig 1 results: Figure 1 only reports results on three models from the Gemma family and Mistral. It is unclear why common benchmarks such as GPT, Claude, or Gemini are omitted. Broader model coverage would strengthen generality claims.

8. Single dataset limitation: Experiments rely on a single dataset, limiting claims of robustness and generalization. Bias evaluation across multiple hiring datasets or tasks would make conclusions more credible.

**Questions:**

1. The paper reports “anti-stereotyping” effects; that is, cases where Black or female candidates are preferred over white or male candidates. What is the occupation distribution in these results? Are the occupations predominantly counter-stereotypical (e.g., women in male-dominated fields)? This finding requires clearer interpretation, ideally supported by a breakdown of results by occupation category.

2. What do the authors mean by this: “whitening” demographic directions?

3. The authors state that “RL-trained reasoning models may exhibit more faithful CoT, but we found no evidence of this.” However, reasoning models are not evaluated in the paper. This claim lacks empirical grounding and should be removed or should be further supported.

---

> ### Author Response · Authors · 2025-11-18
>
> We thank the reviewer for their detailed and constructive feedback. We acknowledge that the reviewer has identified several valuable issues with our presentation and framing, which we will address in our revision. Below we respond to each concern:
>
> **W1: Overstatement of realism**
>
> We agree and have rephrased our claims in our contributions. Our core contribution is demonstrating that adding contextual nuances to existing bias evaluations from the literature, which found no bias with anti-bias prompts, can induce significant and consistent bias. We acknowledge this does not constitute fully "realistic" hiring scenarios and have adjusted our language accordingly.
>
> We do note that our company culture blurbs were manually extracted from public career pages (URLs in Appendix K).
>
> **W2: Prompt debiasing claims conflict with prior work**
>
> We appreciate this point. As discussed in our related work (though we have made this more prominent), recent work (Wang et al. https://aclanthology.org/2024.findings-emnlp.184/ and An et al. https://academic.oup.com/pnasnexus/article/4/3/pgaf089/8071848?login=false) has found that prompt-based debiasing sometimes fails to prevent bias, with conflicting findings in the literature. Our emphasis is specifically on examining previous work that found no bias and testing whether these results hold when additional context is introduced. We will clarify this framing.
>
> **W3: Missing prompt details**
>
> All prompts, including the four debiasing prompts, are included in Appendix K.
>
> **W4: Insufficient baselines**
>
> We agree that testing stronger prompt baselines would be valuable. However, we note that across six models (GPT-4o, Claude 4 Sonnet, Gemini 2.5 Flash, Gemma-2 27B, Gemma-3, Mistral-24B), we find consistent bias across all four anti-bias prompts in many realistic settings. This suggests it may be challenging to find prompts that work robustly across all models and settings, whereas our internal intervention achieves this consistently. We acknowledge this limitation and view testing methods like iterative self-debiasing as valuable future work.
>
> That said, our core finding is that the anti-bias prompts that were reported as effective in Veldanda et al. and Tamkin et al. (https://arxiv.org/abs/2312.03689) when evaluated in simple settings are not actually effective when evaluated in settings with additional contextual details. This concretely highlights the importance of more realistic evaluations, a point that remains true even if other more complicated prompting strategies might exist.
>
> **W5: Loose use of interpretability**
>
> We agree and have modified our claims in our updated manuscript. Our method is more accurately described as an internals-based intervention.
>
> **W6: Choice of solution requires justification**
>
> We agree. To be clear, we are not proposing a new method, we instead view ourselves as applying a standard directional ablation approach that has been used in the interpretability literature. This technique performs very well on our evaluations, and after intervention, our bias measurements are statistically indistinguishable from zero in most cases (95% confidence intervals include zero). We did not investigate more complex methods like FairSteer because this simple approach was highly effective.
> We have clarified that the method itself is not original, only the finding that it is more effective than prompting baselines at reducing bias in hiring.
>
> **W7: Limited Fig 1 results**
>
> We include GPT-4o, Claude 4 Sonnet, and Gemini 2.5 Flash in Figure 2. They are excluded from Figure 1 only because we lack weight access to these commercial models, preventing us from testing internal interventions.
>
> **W8: Single dataset limitation**
>
> We agree. While we evaluate across a range of prompts, company contexts, and six diverse models with consistent results, additional datasets would strengthen our claims. We view this as valuable future work.
>
> **Q1: Anti-stereotyping effects and occupation distribution**
>
> We evaluate across the IT occupation category, which is predominantly white and male. The observed preference for Black and female candidates is therefore counter-stereotypical, which may influence this pattern.
>
> **Q2: What is "whitening"?**
>
> Whitening refers to scaling the demographic direction by the element-wise standard deviation of activations (Equation in Methods, Step 2). This normalization helps account for varying activation magnitudes across dimensions and was particularly helpful for reducing MMLU degradation on Gemma-3 models. Please see https://en.wikipedia.org/wiki/Whitening_transformation for more information.
>
> **Q3: Claim about RL-trained reasoning models**
>
> As stated in our results section, we evaluate Claude 4 Sonnet's reasoning traces, which are trained with reinforcement learning.

---

> > ### Comment · Reviewer_TqmZ · 2025-11-18
> >
> > I appreciate the authors attempt to improve / address review feedback. I acknowledge that I have fully read author responses to this and other reviews. I believe the authors will need to reframe the paper in significant ways to address feedback provided. This tied together with the absence of novel methodology, I will retain my original score.

---

### Official Review · Reviewer_aZFN · 2025-10-31

**Soundness:** 1
**Presentation:** 1
**Contribution:** 1
**Rating:** 2
**Confidence:** 4

**Summary:**

The authors study the fairness of LLMs in the context of screening candidate resumes. The paper concludes that, at first glance, there is little (but significant) bias from all the open source and commercial models examined, and this bias can be successfully mitigated using simple anti-bias prompts. However, when the tasks is enriched by providing additional description about the hiring company culture (from the company website) or requesting selective criteria (e.g., only top 10%) suddenly all models exhibit stronger bias towards preferring female over male and black over white candidates. Furthermore, in this scenario the authors claim that simple anti-bias prompt are no longer effective. Instead the authors tried an existing technique (affine concept editing - ACE  from Marshall et al., 2025) for concept editing based on identifying directions in activation space correlated with demographics and at inference time shift activations along those directions to a neutral midpoint. This approach resulted in a more effective bias mitigation than prompting while minimally impacting model performance (measured as MMLU).

**Strengths:**

- The call for action to use more realistic and challenging evaluation for bias is very important.
- The overall raising awareness about bias in LLMs, especially when employed for important decision such as hiring is also very important.
- The authors examine a good set of models both commercial and open sourced.
- The proposed intervention seemed effective on an individual demographic axes.

**Weaknesses:**

The paper mentions that biases favor Black over White and female over male in their setting — this is counter to many fairness concerns (which typically focus on disadvantage to historically marginalized groups). When such strong claims are presented the evaluation protocol needs to be extremely solid, clearly explained and results need to be thoroughly analyzed. While this should be true always, in this specific context, it is even more important. Unfortunately I find the paper lacking in all those aspects.

Protocol for the bias evaluation
- Lack of control for confounders. From the paper it is unclear how the exact task / evaluation was carried out. Specifically, there is the risk that the model is choosing a candidate not because of race or gender but because the resume could have been objectively better. The authors argue that this is not the case because bias is reduced after the concept edit of those demographics, however, this remains an indirect proof and other factors may play a role when one starts to modify the model’s activations. In an extreme example, if I make the decision completely random bias will go away, yet it does not mean that the model was using race/gender to make the decision. Please understand this is just an extreme example to drive my point home. A standard fairness protocol would hold all other features constant and vary only the sensitive attribute. In my opinion the correct way to assess this would be to take the same resume and create two copies, modifying those cues that might induce the model to believe the come from different gender and race candidate and then measure if the model still have a preference. It is not fully clear whether this “counterfactual” style control was achieved, and therefore, why this bias is being measured.
- One of the main finding is that biases in “realistic” contexts favor Black over White and female over male. The authors proposed some hypothesis about why this might emerge but the protocol does not deeply explore why the bias flips/arise in this direction. With such strong claims one needs a strong protocol (as per my point above) and a deeper analysis of the results to understand what is happening.

Protocol for the mitigation
- The authors correctly emphasize the importance of realistic tasks. However, the test they conduct analyze independently per axis (either gender OR race) while in practice those axis appear in conjunction. This is important for two aspects: on the one hand intersectional biases tend to be even stronger than single axes biases, on the other hand multi-concept editing is known to be particularly challenging. So there is no evidence that the proposed mitigation, in real tasks scenarios, will actually work. Given the emphasis posed on the realistic tasks this is a rather strong unrealistic simplification.
- Prompt results are aggregated across all prompts tested. Why? The most interesting results is for each model the result from the best prompt. In fact from the results in the appendix it seems that for every model there exists one prompt that reach 0 bias in both race and gender.
- The authors focus on Chain of Thought (CoT) but there is no example of few shots which might be more effective. Providing a specific resume where cues injected to imply different gender/race and instruct the model that those two resumes should be valued equally (this is just an example) might actually be more effective than CoT.
- Some results are difficult to compare. For example Table 5 shows that some prompts seem effective for the acceptance rate task in the “realistic scenario” with additional context from Meta. However the comparable results using the proposed mitigation are not presented.
- [Minor] While I believe that few shot might work better than CoT, the CoT evaluation seems limited: manual keyword search + GPT-4o automated review of reasoning. The exact methodology for evaluating faithfulness is not fully described (what keywords, what thresholds, how many samples). This part is not clear and rigorous and may benefit from more formal probing.


Overstatement / not fair presentation of the results / weak arguments
- In some cases the concept editing mitigation seems to be overcorrecting. To the extent that it ends up inducing a pro-stereotypical bias of a comparable, or at times even higher magnitude. E.g., Figure 3 (a) Gemma-3 12B and Mistral,  Figure 5 Gemma-2 27B, Figure 7 Gemma-3 12B. Yet these weaknesses are not commented.
- Some claims are overstated. For example “biases consistently favor Black over White candidates and female over male candidates across all tested models and scenarios” is strong, yet there are example in the Appendix (e.g., Figure 5 and 6) where the bias is in the opposite direction. It would help to tone the statement down given those results.
- Some arguments seem counter intuitive and weak. For example the authors state that “Internal interventions have intuitive advantages over external methods such as prompting. Real-world hiring contexts are inherently complex and multifaceted, involving countless variations of job descriptions, domains, prompts, and candidate information. Ensuring consistently unbiased responses across every possible input scenario via prompt engineering alone may be unrealistic.”  Surely changing the prompt is much more intuitive and easier to do than performing concept editing. Additionally, a “ complex and multifaceted” task surely requires multi-concept editing which, as mentioned above, the current empirical evaluation do not present (and multi concept editing is known to be challenging).

Writing
- Sampling / dataset construction: The setup uses synthetic or canned “candidate resumes” and “job descriptions” plus added context. But the paper gives limited detail  about how the candidate/resume set was constructed (number of candidates, how demographic attributes were randomized, how subtle cues like “College affiliation” were inserted). While Appendix provides some, but not all, this info the main paper could better summarize it.
- Some of the writing can be controversial and lack adequate citation and support. For example “To signal White candidates, we used the predominantly White institutions (PWIs) Georgetown University and Emory University.” According to who?
- The paper report standard deviation but it is unclear out of how many runs and what was different in each of them.
- The exact way the intervention for concept mitigation was constructed is not clear. The authors state “Crucially, our intervention (constructed from explicit name-based demo-graphic signals) successfully mitigates these implicit biases.” But how the synthetic dataset was used is missing.
- Table 1 shows the MMLU when intervening (I suppose) on race (it’s actually not fully clear). But the equivalent table when intervening on gender is not presented.
- [Minor] Many key information are in the appendix. For example the exact prompts used appear there but are not referenced from the main paper. In section 3.2 when talking about prompting you should add the reference to the Appendix where the actual prompts are listed.


Suggestions
- You have some hypothesis that adding “only accept candidates in the top 10%” etc… causes the bias to appear but there is no ablation that shows which of these contextual information induce the bias. This would be a great addition to the work.
- In addition to MMLU it would be great to add Perplexity and some measure of fluency.
- As mentioned above I don’t believe CoT would be particularly useful. However, an interesting experiments could be to test CoT before and after concept editing and analyze the differences in the CoT.
- The full work assumes that demographic fairness equates to simply balancing acceptance rates by race/gender. This is a perhaps understandable simplification yet this aspect should be discussed in the limitation section (and maybe even mentioned in the introduction).

**Questions:**

Unfortunately all the points above would need to be addressed and I don't believe this is doable within a rebuttal. Even assuming that the protocol is correct and it is simply a matter of better explanation that would completely change the manuscript and would require a new review. However, I remain open to the fact that I might completely miss understood the work and would be happy to discuss it with the authors.

During the rebuttal please don’t worry about all the points tagged as “Minor” nor the Suggestions. Those are for you to consider in order to make the work stronger.

**Details Of Ethics Concerns:**

The paper concludes that biases favor Black over White and female over male in their setting — this is counter to many fairness concerns (which typically focus on disadvantage to historically marginalized groups). However such strong conclusions are not supported by a strong and clear protocol evaluation.

I am not sure this is "just" a reason to reject or if Ethics review needs to be carried out.

---

> ### Author Response · Authors · 2025-11-18
>
> We thank the reviewer for their detailed feedback. We believe several of their primary concerns stem from insufficient clarity in our presentation rather than fundamental methodological issues. Specifically, our experiments already employ the counterfactual evaluation and joint-axis interventions the reviewer requests (W1, W2). Below we clarify these points and address how we have improved the manuscript.
>
> **“Lack of control for confounders”.**
>
> The reviewer asks for a counterfactual setup where “the same resume \[has\] two copies, modifying those cues that might induce the model to believe they come from different gender and race candidates.” This is exactly the protocol we use.
>
> Our hiring experiments use the framework, dataset, and code directly from Veldanda et al. (2023, https://openreview.net/pdf?id=erl90pLIH0). Each resume is paired with four different names (White male, White female, Black male, Black female), producing counterfactual variants that are identical except for the name that signals demographics. Our main modification is to add contextual information (company culture text, top 10% constraint) to the prompt while keeping the underlying resume text fixed. We have made this counterfactual construction much more explicit in the main text.
>
> **“the test they conduct analyze independently per axis (either gender OR race) while in practice those axis appear in conjunction”**
>
> The reviewer states that we “analyze independently per axis (either gender OR race).” While our plots report race and gender gaps on separate panels for readability, the underlying experiments always vary both attributes jointly, and the intervention is always applied to both race and gender directions simultaneously. We have clarified this in the methods section.
>
> **“from the results in the appendix it seems that for every model there exists one prompt that reach 0 bias in both race and gender.”**
>
> The reviewer notes that “from the results in the appendix it seems that for every model there exists one prompt that reaches 0 bias in both race and gender.” We were not able to identify a realistic setting in our experiments where a single anti bias prompt produces bias estimates that are clearly indistinguishable from zero for both race and gender. If the reviewer could specify which table/figure they are referring to, we would be happy to clarify.
>
> **“Prompt results are aggregated across all prompts tested. Why? The most interesting results is for each model the result from the best prompt.”**
>
> We aggregate across the four anti-bias prompts to evaluate robustness: all four were effective in existing simple settings from the literature, and we test whether this effectiveness generalizes to more realistic settings where e.g. we add information about the company. Across six models and four prompts in more realistic settings, we consistently observe statistically significant bias, suggesting that prompt-based mitigation is unreliable when realistic details are added.
>
> It is true that more sophisticated prompting strategies than studied in the prior literature might be able to reduce bias. That would be a valuable direction for future work.
>
> That said, our core finding is that the anti-bias prompts that were reported as effective in Veldanda et al. and Tamkin et al. (https://arxiv.org/abs/2312.03689) when evaluated in simple settings are not actually effective when evaluated in more realistic settings. This concretely highlights the importance of realistic evaluation, a point that remains true even if other more complicated prompting strategies might exist.
>
> **“The authors focus on Chain of Thought (CoT) but there is no example of few shots which might be more effective.”**
>
> We follow the established zero shot / anti bias prompting methodology of Veldanda et al. Adding few shot examples or more complex prompting schemes would introduce additional variables beyond those prior setups and make it harder to interpret differences. We agree that exploring richer prompting strategies (few shot, self debiasing, etc.) is valuable future work.
>
> **“For example Table 5 shows that some prompts seem effective … However the comparable results using the proposed mitigation are not presented.”**
>
> The reviewer notes that Table 5 lacks comparable mitigation results. These are provided in Table 6, which reports the corresponding numbers after applying our internal intervention.
>
> **“In some cases the concept editing mitigation seems to be overcorrecting.”**
>
> The reviewer correctly notes residual bias in some cases after intervention. We will clarify that: (1) pre-intervention bias reaches up to 11%, (2) post-intervention bias is always below 2.5% and typically under 1%, (3) in almost all cases, post-intervention bias is not statistically significant, and (4) in cases where the bias sign flips, this residual bias is never statistically significant. We have added explicit discussion of this pattern.

---

> > ### Author Response · Authors · 2025-11-18
> >
> > **“Some claims are overstated. For example “biases consistently favor Black over White candidates and female over male candidates across all tested models and scenarios” is strong, yet there are example in the Appendix (e.g., Figure 5 and 6\) where the bias is in the opposite direction.”**
> >
> > We have clarified our claim about bias direction to be more precise. In our experiments, when the measured bias is statistically significant (confidence intervals exclude zero), the direction consistently favors Black over White and female over male within our tested settings. When bias is small and confidence intervals include zero, the sign can vary, as the reviewer correctly notes.
> >
> > We have revised the abstract and introduction to state: "When these biases emerge and are statistically significant, they consistently favor Black over White candidates and female over male candidates across all tested models and scenarios." This more precisely captures our empirical findings without overstating the claim.
> >
> > **“changing the prompt is much more intuitive and easier to do than performing concept editing.”**
> >
> > We agree that modifying the prompt is operationally simpler and more intuitive than editing internal activations. However, our main empirical finding is that, across six models and four anti bias prompts (including ones from prior work), prompt based mitigations are not effective in more realistic settings (even if they appear to be effective when evaluated in simple settings). This justifies the need for an alternative mitigation, even one that comes at the cost of being more complex.
> >
> > **Clarifications on writing and methodological details**
> >
> > We appreciate the reviewer’s comments on missing or underspecified details and will address them as follows:
> >
> > * **“Sampling / dataset construction”:** We will clearly state that we:
> >
> >   * Use resumes and names directly from the public dataset of Veldanda et al. (2023), with four names per resume corresponding to the {White/Black} × {male/female} combinations, and
> >
> >   * Use the synthetic templates and name based demographic signals from Tamkin et al. (2023) for constructing demographic directions.
> >
> > * **“The paper report standard deviation but it is unclear out of how many runs and what was different in each of them.”** The reported standard deviations and confidence intervals are computed across the four anti bias prompts for each model and setting. That is, the four runs differ only in the debiasing instruction.
> >
> > * **“The exact way the intervention for concept mitigation was constructed is not clear.”** We have expanded the description of the concept editing procedure in the main text: we compute mean activations for each demographic group across all token positions in the synthetic dataset, form whitened difference vectors, normalize them, and apply affine editing to move projections to the midpoint between group means for both race and gender.
> >
> > * **“Table 1 shows the MMLU when intervening (I suppose) on race (it’s actually not fully clear). But the equivalent table when intervening on gender is not presented.”** Table 1 reports MMLU when intervening on both race and gender simultaneously.
> >
> > * **College affiliation and PWI / HBCU designations.** We have replaced 'predominantly White institutions' with ‘colleges which have historically had majority white enrollment:’
> >
> > **“You have some hypothesis that adding “only accept candidates in the top 10%” etc”**
> >
> > The reviewer suggests that we “hypothesize” that the “top 10%” constraint may cause bias. In our results this is a direct empirical observation rather than a hypothesis. Adding a single sentence, such as “only accept candidates in the top 10%,” to the prompt increases measured bias across all tested closed-source models (see Figure 2b for this experimental result).
> >
> > **Ethics concern**
> >
> > We share the reviewer’s view that work in this space must be especially careful. Our evaluation uses established methodologies from peer reviewed work (Veldanda et al., Tamkin et al.), including their released datasets and code. Our main methodological change is to add realistic but simple contextual information to the prompts (company culture text and a selectivity constraint). Across all six models, this small change is sufficient to cause substantial bias that was not apparent in the original evaluations, and we view documenting this fragility as an important contribution. Given this unanimous finding using a methodology taken directly from prior work with only the minimal modification of adding simple contextual information about the company, we find it hard to escape the conclusion of bias in the direction we report.
> >
> > Other researchers have also reported similar bias directions in prior work, such as Wang et al. ([https://aclanthology.org/2024.findings-emnlp.184/](https://aclanthology.org/2024.findings-emnlp.184/))  who found that “Seven out of ten LLMs show significant biases against males in at least one industry” in hiring scenarios.

---

> ### Comment · Reviewer_aZFN · 2025-11-19
> **Thank you for your rebuttal**
>
> Thank you for providing more explanation. I could initiate a discussion about each individual point but the exact reason that so much clarification is needed to me is sufficient to say that the paper, at the very least, needs to be rewritten in a way that is more clear and being reviewed again.
>
> Due to the lack or confusing information provided I am still unsure about the methodology. I still think that prompts should not be averaged, you should report the results of the most effective prompts (otherwise it's like averaging the results of an algorithm using different hyperparameters rather than the most effective ones). I still think that multi-shot rather than CoT should be tested (independently of what previous work might have done).
>
> I believe this kind of work is very important so I hope these authors will persist and that this review can help them to improve their work and their manuscripts for future submissions.

---

### Meta-Review · Area_Chair_EueH · 2025-12-23

**Summary:**

The paper investigates the fairness of LLMs for recruiting decisions, revealing that prompt-based bias mitigation is insufficient in realistic scenario where a more detailed context is provided. The authors further apply activation steering, which is widely adopted in interpretability literature, to alleviate the biases in this case. The reviewers have reached an agreement on some concerns of the submission, e.g., overclaim without in-depth analysis, indicating the necessity of a major revision. Besides, the proposed internal manipulation method can be applied merely on open-sourced LLMs, limiting its real-world applicability. Therefore, my recommendation at this point is to reject.

**Reviewer Concerns:**

The reviewers pointed out the following concerns regarding the submission.

- aZFN: unclear evaluation protocol (addressed); overclaim that requires deeper analysis; insufficient fairness evaluation, i.e., lack of intersectional case; overstatement in experiment results; and some writing issues.

- TqmZ: overstatement for evaluation setting; missing prompt details; lack of justification of methodology; insufficient baseline and dataset.

- Y857: insufficient mechanistic clarity and design choices; there is a need for more ablations and analysis results.

Even though the authors have addressed some of the aforementioned weaknesses by rephrasing the claims and adjusting the manuscript, it still needs to be greatly reframed for acceptance.

**Reviewer Scores:**

Reviewers aZFN and TqmZ would keep their original scores, while Y857 may tend to decrease to a negative rating.

---

### Decision · Program_Chairs · 2026-01-26

Reject